# Incidence of non-Hodgkin's lymphoma among adults in Sardinia, Italy

**Giorgio Broccia**[1], **Jonathan Carter**[2], **Cansu Ozsin-Ozler**[3], **Federico Meloni**[4], **Sara De Matteis**[4], **Pierluigi Cocco**[4,5] *

**1** Department of Haematology and Bone Marrow Transplants, Hospital A. Businco, Cagliari, Italy,
**2** University of Coventry, Coventry, United Kingdom, **3** Department of Paediatric Dentistry, Faculty of Dentistry, Hacettepe University, Ankara, Turkey, **4** Department of Medical Sciences and Public Health, University of Cagliari, Monserrato, Italy, **5** Centre for Occupational and Environmental Health, Division of Population Health, University of Manchester, Manchester, United Kingdom

* pcocco@unica.it

## Abstract

The causes of the peculiar time trend in the incidence of non-Hodgkin's lymphoma (NHL) in most parts of the world and of its geographic distribution are still unknown. We used the data base of 1974–2003 incident cases of hematological malignancies to explore the time trend of NHL incidence in the region of Sardinia, Italy, and we used Bayesian methods to plot the probability of NHL incidence by residential unit on the regional map. In 1974–2003, 4109 NHL cases were diagnosed among resident adults in Sardinia, with an incidence rate of $13.38 \times 10^{-5}$ (95% CI 12.97–13.80). NHL incidence showed an upward trend along the study period with an average annual percent change (APC) of 4.94 (95% CI -5.39–16.4), which did not vary by gender or by age-group. Cancer registry data, covering part of the region starting from 1993, suggest that the increasing trend did not persist in the subsequent years. Areas with the highest probability of an excess incidence tended to cluster in the north-eastern part of the region and in two major urban centers, with the low incidence areas located in the south, confirming previous observations. Prevalence of viral infections, environmental and occupational exposures, or socio-economic deprivation would not explain the peculiar geographic distribution we observed. These findings provide convincing arguments for extending the coverage of routine cancer registration over the whole Sardinian population, while prompting further research on the genetic and environmental determinants of NHL in the risk areas.

## Introduction

After a steady increase for several decades worldwide, in most parts of the World, incidence of non-Hodgkin's lymphoma (NHL) reached the top of the curve around the 1990's and stabilized thereafter [1]. According to the latest estimates from the 10[th] Edition of the IARC Cancer Incidence in the Five Continents, across the 33 Italian Cancer Registries, NHL incidence ranges $9.7–17.4 \times 10^{-5}$ among males and $6.5–13.4 \times 10^{-5}$ among females [2], and it differs between the two Cancer Registries operating in the island of Sardinia, a region of Italy. In fact,

**Data Availability Statement:** All relevant data are within the manuscript and its Supporting Information files.

**Funding:** This research did not receive any grant from funding agencies in the public, commercial, or not-for-profit sectors.

**Competing interests:** The authors have declared that no competing interests exist.

the standardized incidence (world population) in 2008–2012 was 12.1 among males and 8.6 x $10^{-5}$ among females in the Sassari Cancer Registry, and 17.4 among males (the highest in Italy) and 11.2 x $10^{-5}$ among females in the Nuoro Cancer Registry. The Sassari Cancer Registry covered the northern part of the island, including one-third of the regional population, from 1993 and it was discontinued from 2012. The Nuoro Cancer registry started operating in 2003, and it covers the central-eastern part of the island and only 13% of the regional population. A cancer registry has been recently planned in southern Sardinia as well, but its implementation is not easily foreseeable. Previous reports showed a peculiar pattern of the geographic distribution of NHL mortality [3] and incidence [4] over the territory of Sardinia, with high mortality and incidence areas clustering mainly in the central and northern part of the island. However, the lack of a uniform, stable routine system of surveillance of cancer incidence, covering the whole territory of Sardinia, has been a major impediment in exploring long term time trends and the geographic distribution of cancer. Such information would be of paramount relevance for a cost-effective planning of the cancer care resources, in terms of location, equipment and staff. Besides, the knowledge of the time and space coordinates of disease is basic to elaborate hypotheses and to address proper investigations on its determinants.

The population of Sardinia is well known among geneticists for its peculiarities [5]. Due to a millennial geographic isolation and selective pressure from malaria, gene polymorphisms associated with a prevalence of multiple sclerosis [6], childhood type I diabetes [7], thalassemia [8], and other rare diseases have become highly prevalent. Also, specific areas of the island have one of the highest concentration of male centenarians worldwide [9]. Extensive research has investigated the multiple genetic traits associated to NHL, but these would explain only a fraction of the incident cases [10]. Therefore, the study of the time trend and the geographical distribution of the NHL cases among the genetically peculiar Sardinian population might provide clues in this regard.

To compensate for the lack of official records, the chief hematologist of the Cagliari Oncology Hospital (BG) initiated and regularly updated a database of incident cases of all hemolymphatic malignancies in Sardinia from 1974–2003, with the collaboration of all the clinical, surgical and pathology departments, social security agencies, and health authorities of the island [11]. Validity and completeness of the records were previously validated by comparison with mortality and hospitalization data [12]. We used this database to explore the time trend and the spatial distribution of NHL risk by residential units over the whole island.

## Material and methods

### Study population

The database of hemolymphatic malignancies we used in this study was described in detail elsewhere [11]. Briefly, it includes all incident cases of any hematological cancer, diagnosed among the Sardinian population of both genders, and at any age in 1974–2003. For the purposes of this analysis, we selected the 4109 NHL cases occurring among subjects aged 25 year or more, including 1875 women and 2234 men. The available records include gender, 10-year age-group, residence, oncohematological diagnosis, and year of diagnosis of 14,744 patients. All records were anonymous.

### Statistical methods

The total population of Sardinia increased from 1.5 to 1.62 million along the study period. The 1971, 1981, 1991, and 2001 census population of the 356 communes of the island was extended four years onwards and five years backwards to estimate the resident adult population in the intercensal years by gender, and 10-years age group (25–34, 35–44, 45–54, 55–64, 65–74, and

75+). For each commune, the smallest administrative unit in Italy, the standardized annual incidence rate of NHL along the study period was calculated by applying the regional rates over the study period to the person-years in the corresponding strata of the local population. The time trend of the regional annual incidence rate in 1974–2003 was calculated with the best fitting regression equation, as determined by the $R^2$ value. The null hypothesis of no time variation was tested with the Pearson's correlation coefficient, and it was rejected if associated with a probability $< 0.05$.

The geographical distributions of the probability of NHL incidence rate across the 356 communes existing in Sardinia in 1974 was explored using a Bayesian approach [13] on the map of their territorial borders. The map contains information from the Italian Institute for Statistics (ISTAT) (https://www.istat.it/it/archivio/104317), which is made available under the Creative Commons BY 3.0 IT License. Another 21 communes were created subsequently during the study period by separation from the original administrative unit, which we incorporated in the original commune so to preserve consistency across the time.

We first calculated the posterior probability of the NHL incidence rate η in each commune through the following equation:

$$P(\eta|d, I) = \frac{P(d|\eta, I)P(\eta|I)}{P(d|I)} \tag{1}$$

where $P(\eta|d,I)$ is the posterior probability distribution of the NHL incidence rate η for an individual commune. $P(\eta|I)$ is the prior standardized NHL incidence rate, η, based on the regional data $I$, and $P(d|\eta, I)$ is the likelihood function, i.e. the probability of getting the number of cases $d$ for a given commune. The term $P(d|I)$ does not depend on $\eta$; it ensures that $P(\eta|d, I)$ is a properly defined probability density function.

The diagnosis of NHL in an individual in a given year is a binomial process. Therefore, the number $k$ of cases observed in a population of size $n$, measured in person-years, is binomially distributed. Since $k \ll n$, we can approximate it with the Poisson distribution:

$$P(d|\eta, I) = P(k, n|\eta, I) = e^{-n\eta} \frac{(n\eta)^k}{k!} \tag{2}$$

where $n\eta$ is the expected number of cases in a population of size $n$. Once established the number of cases and the population size for each commune, it is possible to calculate the posterior probability of the observed number of cases for the gender and age subgroups within each commune. To identify those communes which posterior probability is located on the extreme right of the curve of the distribution of the incidence rate of the 356 communes in the region, we arbitrarily defined a critical threshold, that we set at 0.999, i.e.

$$\text{Prob}(\eta < \eta_c|D, I) = \int_0^{\eta_c} P(\eta|D, I) \, d\eta = 0.999 \tag{3}$$

where $D$ is the data for the whole region. The critical thresholds for each age and gender group at regional level are given in Table 1.

Finally, the probability $H1$, that the incidence rate in a commune is higher than the critical incident rate for the whole region is given by:

$$P(H1|d, I) = \int_{\eta_c}^1 P(\eta|d, I) d\eta \tag{4}$$

We used bespoke python code to calculate the probability of exceeding the critical NHL incidence rate. The ratio between the probability associated with H1 *vs* that associated with H0 (the null hypothesis of chance variation) is the likelihood ratio, which follows its own probability distribution. We plotted such probability on the regional map using the following color

**Table 1. Critical thresholds $\eta_c$ used to calculate the likelihood ratio for each group in the fourth data set.**

| Age and gender study group | Critical value of $\eta$ ($\eta_c$) |
|---|---|
| 25–34 Females | 0.0000355 |
| 35–44 Females | 0.0000595 |
| 45–54 Females | 0.0001140 |
| 55–64 Females | 0.0001740 |
| 65–74 Females | 0.0002940 |
| 75+ Females | 0.0002705 |
| 25+ Female | 0.0000805 |
| 25–34 Males | 0.0000535 |
| 35–44 Males | 0.0000905 |
| 45–54 Males | 0.0001290 |
| 55–64 Males | 0.0002325 |
| 65–74 Males | 0.0003560 |
| 75+ Males | 0.0003735 |
| 25+ Males | 0.0000970 |
| 25–34 Total population | 0.0000410 |
| 35–44 Total population | 0.0000705 |
| 45–54 Total population | 0.0001155 |
| 55–64 Total population | 0.0001935 |
| 65–74 Total population | 0.0003100 |
| 75+ Total population | 0.0002985 |
| 25+ Total population | 0.0000870 |

scale for the area of each commune, based on the quintiles of its distribution: white ≤ 0.165, light grey 0.166–0.335, medium-light grey 0.336–0.50, medium-dark grey 0.501–0.80, dark grey 0.801–0.95. The few communes associated with a probability higher than 95% had a black shade.

No patient or public was involved in this study. The use of the database of hemolymphatic malignancies diagnosed in 1974–2003 among the Sardinian population for the purposes of scientific publication was approved by the Ethics Committee of the University Hospital of Cagliari (protocol N. PG 2019/18070, 18 December 2019), in agreement with the Code of Ethics of the World Medical Association (Declaration of Helsinki).

## Results and discussion

### Time trend in NHL incidence

In 1974–2003, 4109 NHL cases were diagnosed among subjects aged 25 years or older resident in Sardinia. The incidence rate was 13.38 x $10^{-5}$ (95% CI 12.97–13.80) for the total population, 15.09 (95% CI 14.47–15.73) among men, and 11.79 (95% CI 11.26–12.34) among women; the male/ female ratio was 1.19, based on the actual number of cases.

Graphs in Fig 1 show the trend of NHL over the 30-year period, by gender. Incidence of NHL increased linearly by 1.33 x $10^{-5}$ per year on average along the study period ($p$ = 8.23 x $10^{-15}$) (annual percent change [APC]: 4.94, 95% CI -5.39–16.4) (Table 2). The increasing trend was observed likewise among the female (1.134 x $10^{-5}$ per year, $p$ = 5.36 x $10^{-13}$; APC: 5.10, 95% CI -5.23–16.55) and the male population (1.600 x $10^{-5}$ per year, $p$ = 7.20 x $10^{-11}$; APC: 4.38, 95% CI -4.99–14.68), as well as in all age strata (age 25–44: APC: 4.95, 95% CI -6.49– 17.71; age 45–64: APC: 4.27, 95% CI -12.06–23.67; age 65+: APC: 5.27, 95% CI -5.88–17.65).

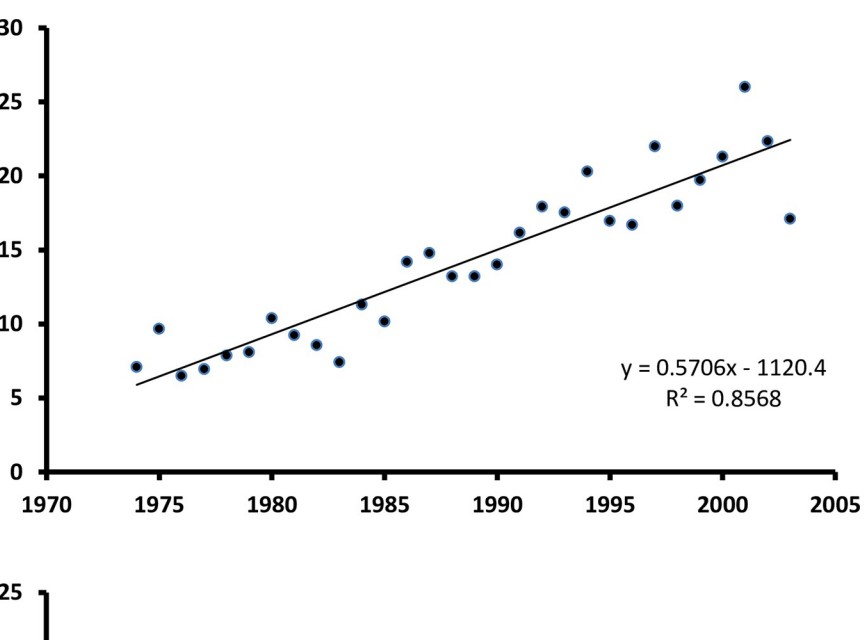

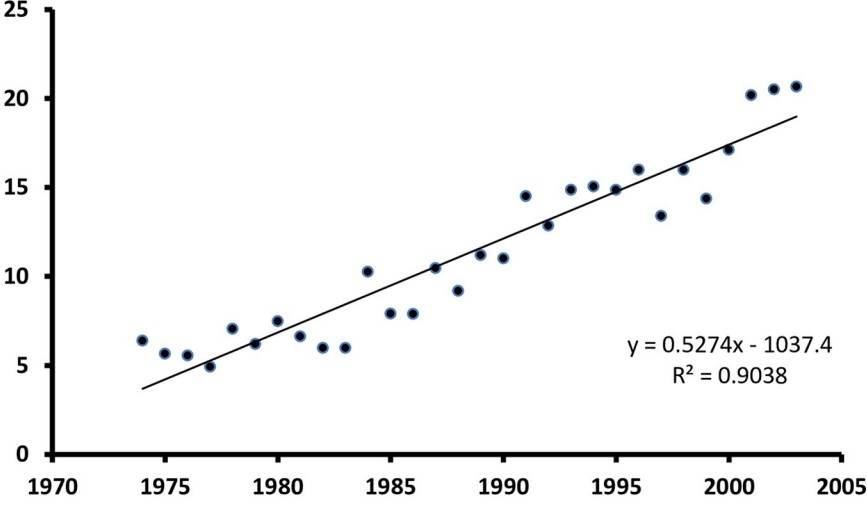

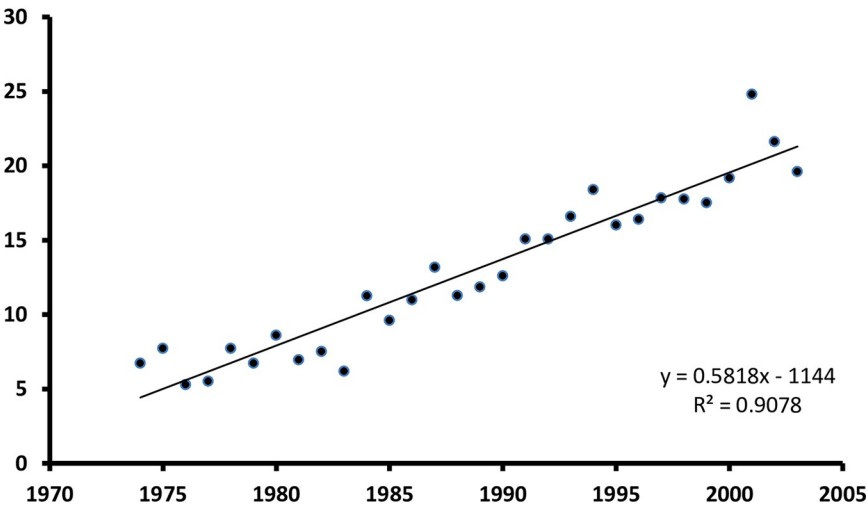

**Fig 1.** 1974–2003 annual incidence rate of non-Hodgkin's Lymphoma in Sardinia, Italy, by gender (A, B) and overall (C).

**Table 2. Linear regression coefficients describing the time trend of NHL incidence by age group and gender.**

| Age/gender group | β | constant | Pearson's correlation coefficient | p-value |
|---|---|---|---|---|
| All ages | | | | |
| men | 0.5706 | -1120.4 | 0.926 | $2.35 \times 10^{-13}$ |
| women | 0.5274 | -1037.4 | 0.951 | $9.11 \times 10^{-16}$ |
| total | 0.5818 | -1144.0 | 0.953 | $5.01 \times 10^{-16}$ |
| 25–44 years | | | | |
| men | 0.2667 | -524.41 | 0.786 | $2.71 \times 10^{-7}$ |
| women | 0.1775 | -349.15 | 0.818 | $3.50 \times 10^{-8}$ |
| total | 0.2223 | -437.12 | 0.892 | $3.52 \times 10^{-11}$ |
| 45–64 years | | | | |
| men | 0.5885 | -1154.1 | 0.750 | $1.82 \times 10^{-6}$ |
| women | 0.5056 | -993.15 | 0.878 | $1.88 \times 10^{-10}$ |
| total | 0.5465 | -1072.6 | 0.869 | $4.64 \times 10^{-10}$ |
| ≥ 65 years | | | | |
| men | 1.6003 | -3149.0 | 0.886 | $7.20 \times 10^{-11}$ |
| women | 1.1344 | -2230.5 | 0.921 | $5.36 \times 10^{-13}$ |
| total | 3298 | -2615.6 | 0.942 | $8.32 \times 10^{-15}$ |

The regression lines in Fig 1 seem to fit the data efficiently. In fact, after plotting the residuals *vs* the predicted values, we could not detect any knot suggestive of a sudden change in the slope of time-related increase in NHL incidence, which we might have tested with piecewise regression analysis [14]. Therefore, we interpreted the upward trend as continuous along the study period, and not influenced by external events, such as improvements in access to diagnostic facilities, availability of new diagnostic techniques, or known environmental events.

A comparison with Cancer Registry data is possible limited to the northern area of Sardinia, covered by the Sassari Cancer Registry, which started operating and contributing to the IARC Cancer Incidence in the Five Continents (CI5) volumes from 1993. The CI5 volumes report incidence data every 5 years. Therefore, we selected the NHL cases incident in 1998–2002 among the population of any age resident in all communes contributing to the Sassari Cancer Registry and calculated the world population standardized incidence rate by gender, consistent to the IARC standardization procedure. Our figures were $12.9 \times 10^{-5}$ among males and $9.5 \times 10^{-5}$ among females, a little higher than what reported in the CI5 volume IX for the same period (12.1 and $8.0 \times 10^{-5}$, respectively) [15]. By assembling the 1998–2002 NHL incidence data of all communes contributing to the Nuoro Cancer Registry, which started operating from 2003 and contributed to the IARC CI5 from Volume X (2014), the world population standardized incidence rate was $17.4 \times 10^{-5}$ among males and $11.3 \times 10^{-5}$ among females. Based on the Cancer Registry data, the progressive, upward trend we observed in 1974–2003 leveled off in 2003–2007, with standardized rates of 12.8 among males and 10.1 among females in the Sassari Cancer Registry, and 12.9 among males and 10.5 among females in the Nuoro Cancer Registry [16]. The corresponding figures in the IARC CI5 Volume XI for the quinquennium 2008–2012 were 12.1 among males and $8.6 \times 10^{-5}$ among females in the Sassari Cancer Registry, and 17.4 among males and $11.2 \times 10^{-5}$ among females in the Nuoro Cancer Registry [2].

Therefore, our results show that in 1974–2003 NHL incidence increased linearly among the population of the Italian region of Sardinia, in both the female and male population, and in all age groups. Cancer Registry data, partially available from the last decade of the 30-year time span covered by the oncohematology database we used, confirm its validity and completeness

for the years 1998–2002 in northern Sardinia, and suggest no further increase in NHL incidence, consistently with what was observed in most countries worldwide [1].

Taking profit of the local, well-functioning Cancer Registry, another Italian study explored incidence of and survival from peripheral lymphomas, including Hodgkin's and non-Hodgkin's lymphomas and chronic lymphocytic leukemia (CLL), but not multiple myeloma (MM), over a relatively short time span [17]. The authors applied the 2008 WHO classification to identify the individual subtypes and calculated their standardized incidence and survival along the study period. After combining the subtypes included in the NHL category, the observed age standardized (World standard population) incidence in 1997–2003 was $15.6 \times 10^{-5}$ among males, and $12.6 \times 10^{-5}$ among females, higher than our figures for the region of Sardinia in the corresponding period (males: $11.4 \times 10^{-5}$; females: $8.3 \times 10^{-5}$). Unfortunately, the authors estimated the APC indicator for all peripheral lymphomas combined, but not for the individual subtypes [17].

## Spatial variation of NHL incidence

Fig 2 shows the map of the probability that the NHL incidence rate in each Sardinian commune is above the critical rate, overall and by gender.

Eleven communes exceeded the 95% probability of a NHL incidence rate above the critical threshold of its posterior distribution over the whole region. For three of these communes, the

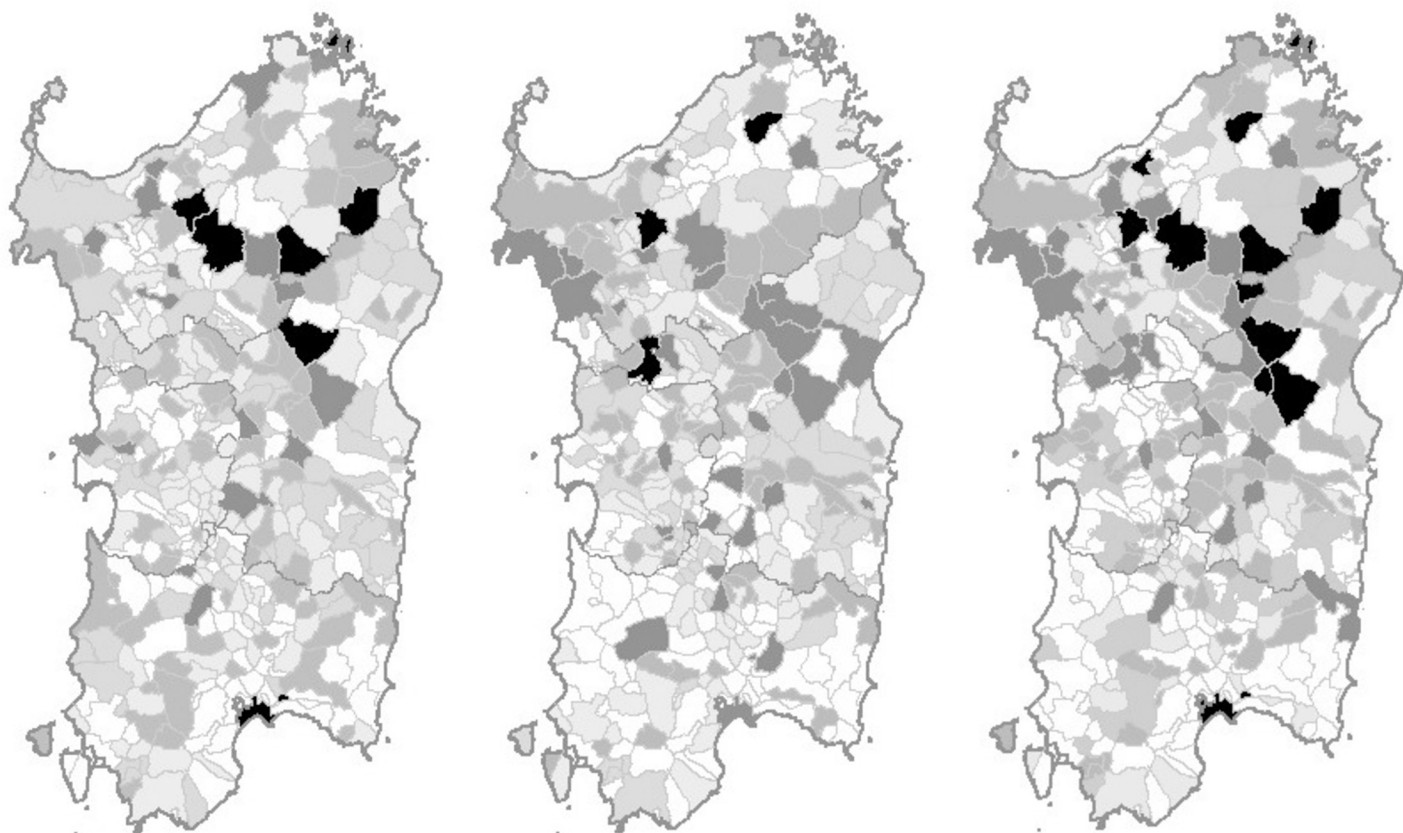

**Fig 2. Map of 1974–2003 NHL incidence in Sardinia, Italy.** The probability of NHL incidence above the critical value is represented by commune with the following color scales: white ≤ 0.165, pale grey 0.166–0.335, pale-medium grey 0.336–0.50, medium-dark grey 0.501–0.80, dark grey 0.801–0.95, black ≥ 0.951. A: males; B: females; C: total population. Maps are available online from the Italian Institute for Statistics (ISTAT) under the Creative Commons BY 3.0 IT license (https://www.istat.it/it/archivio/104317).

excess occurred among the female population only: these are Macomer (21 cases, likelihood ratio 30.7, $p = 0.968$), Luras (9 cases, likelihood ratio 41.4, $p = 0.976$), and Ploaghe (15 cases, likelihood ratio 310.3, $p = 0.997$). For another six communes, there was an excess among the male population only: these communes are Cagliari, the main urban area in the island (321 cases, likelihood ratio 19.1, $p = 0.950$), Nuoro, a town of about 40,000 inhabitants, capital of the central province (60 cases, likelihood ratio 19.7, $p = 0.952$), Buddusò (16 cases, likelihood ratio 22.3, $p = 0.957$), Chiaramonti (8 cases, likelihood ratio 27.1, $p = 0.964$), Ozieri (26 cases, likelihood ratio 27.2, $p = 0.964$), and La Maddalena (28 cases, likelihood ratio 286.6, $p = 0.997$). Finally, there were four communes which excess NHL incidence did show up among the total population only: these are Sedini (10 cases, likelihood ratio 20.1, $p = 0.953$), Orgosolo (22 cases, likelihood ratio 40.7, $p = 0.976$), Mamoiada (10 cases, likelihood ratio 42.8, $p = 0.977$), and Nule (11 cases, likelihood ratio 55.9, $p = 0.982$). Another 22 communes are located on the extreme right of the incidence distribution curve of the 356 communes in the region, with a posterior probability of an excess NHL incidence in the total population ranging 80–94%. On the map, these communes tended to concentrate in the northern (20/35 communes) and central (12/35) areas of the island, with only three in southern Sardinia, including the major urban area of Cagliari.

Eight of the 13 communes, with the highest probability of their rate exceeding that of the overall population of the region in at least one gender, are in the northern area, four in the central area, and one in southern Sardinia.

These findings replicate, extend, and detail the previously observed peculiar pattern of geographic distribution of NHL mortality and incidence [3, 4], which remains largely unexplained. Previous reports, exploring cancer mortality and hospitalization in areas of Sardinia with large industrial and military settlements, observed some excess risk in two out of 8 industrial areas, in one out of three military areas, and in two out of 5 urban areas [18]. Some of the excess risk communes we identified are part of those areas, with the only contrasting finding related to the excess NHL incidence previously reported [4] and herein confirmed for the urban area of Cagliari, which did not show up in the analysis of mortality and hospitalization data [18].

It is well known that viral infections play an important role in the etiology of NHL. Immune depression from HIV infection is perhaps the strongest etiological factor for NHL [19]. However, multicenter studies have shown that other viral infections, such as EBV [20], HHV8 [21], HCV [22], and HBV do play a role [23]. All these viruses might be implicated in the NHL occurrence among the Sardinian population. For instance, HCV infection has been highly prevalent in the Sardinian population, and it has shown an association with NHL risk in a local case-control study [24]. Non-AIDS related Kaposi sarcoma is also especially prevalent among the Sardinian elderly, possibly in relation to their peculiar genetics, although it does not seem to show a clear geographic pattern [25]. Cagliari and its province, but not the provinces of Nuoro, Oristano, and Sassari, was severely affected by the AIDS epidemics in 1989–1993, and paid the highest tribute of lives among the Italian provinces before the introduction of the antiretroviral therapy [26]. Therefore, the AIDS epidemic does not explain the upward trend in NHL incidence herein reported, at all ages and in both genders, and particularly among the elderly, which clustered in the northern and central areas of the island. In the last decades, several studies have highlighted the role of occupational risk factors, such as solvents [27], pesticides [28], and ethylene oxide [29], and environmental contaminants, such as PCBs [30]. None of these would explain the geographic pattern of NHL incidence we observed.

To explore whether social class and lifestyle might have caused the wide geographic variation in NHL across the region, we combined the 356 Sardinian communes into five categories of deprivation index, based on the following socio-economic indicators of the Italian National

Institute of Statistics: proportion of the resident population who attained elementary education at most; proportion of the resident population aged 15 years or older searching for an occupation; proportion of the active population engaged in manual work; proportion of rentals over the total residences; persons per room in the household [31]. We used Poisson regression analysis to calculate the relative risk of NHL for each increasing category of economic deprivation with reference to the lowest. The results showed that NHL risk did not increase by increasing economic deprivation (not shown in the Tables).

Several communes had significantly less NHL cases than expected, including Tortolì (9 observed *vs* 18 expected, RR = 0.5 95% CI 0.26–0.93), Muravera (3 observed *vs* 13.5 expected, RR = 0.2 95% CI 0.08–0.63), and San Vito (2 observed *vs* 13 expected, RR = 0.2 95% CI 0.05–0.52) on the southeastern coast, and Serrenti (3 observed vs 13 expected, RR = 0.2, 95% CI 0.08–0.63) in the central-southern plain. The two towns of Muravera and San Vito are of special interest in this regard, as being very close to each other in a segregated area, surrounded by mountains, the Tyrrhenian Sea and the second longest and largest river of the region on the north. A bridge connecting this area with the opposite bank was built only in 1908. Millennia of isolation might have favored the selection of genes conferring protection against NHL in these towns. We are not aware of genetic investigations aiming to identify the varying prevalence of gene polymorphisms implicated in the pathways leading to NHL development among the Sardinian population. Further studies might provide interesting clues in this regard.

## Limitations

NHL is a complex array of different diseases showing up at different stage of maturation of lymphocytes. NHL classification changed several times along the years the data base we analyzed was being fed, from the Rappaport, Lukes-Collins, and Kiel classifications in the 1970's, to the Working Formulation, the Revised European-American Lymphoma Classification, and the WHO classification and its updates [32]. To preserve the comparability of data collected along three decades, the senior hematologist who reviewed all the diagnoses (GB) maintained the early definition of non-Hodgkin's Lymphoma. This definition includes B-cell and T-cell lymphomas and excludes multiple myeloma and chronic lymphocytic leukemia. Unfortunately, the original slides were not available to reclassify the NHL cases and to explore space distribution and time trends by histological subtypes and grade of malignancy. However, in the previous analysis covering the first two decades (1974–93), the upward trend was observed for both nodal and extra-nodal forms of NHL. This last group includes a greater proportion of Mucosa-Associated Lymphoid Tissue (MALT) lymphomas, which were included among lymphomas only in the late 1980's [33], when the increasing trend was already evident. Besides, the proportion of MALT tumors is highest among the elderly, but the upward trend in NHL incidence occurred in all age groups almost equally. Changes in the therapy protocols, with the introduction of monoclonal antibodies, such as rituximab, have improved the prognosis for most NHL subtypes [34]. This might have been reflected in a mortality study, but not in our study of NHL incidence.

To the best of our knowledge, the spatial distribution of lymphoma subtypes has not been explored thus far. However, thanks to the adoption of the WHO classification, a few Cancer Registry based studies have explored time trends of major B-cell lymphoma subtypes [35–38]. Upward time trends have been consistently described for diffuse large B-cell lymphoma (DLBCL) and follicular lymphoma (FL), two of the most frequent B-cell lymphoma subtypes, while contrasting findings have been reported for CLL [35–38]. As CLL is not included in the NHL definition we adopted, it seems most likely that excess cases of DLBCL or FL or both might have contributed to the increase in NHL incidence we observed in Sardinia in 1974–2003.

A plausible explanation of the clustering of NHL cases might be the special expertise of the local physician/s in detecting the disease, and in correctly addressing the patients to a Hematology department. Such expertise might have increased along the study period as an effect of increased reporting, thus generating a spurious upward trend. However, the increase in NHL incidence we observed was apparently continuous along the thirty years of observation and leveled off afterwards, consistently to what observed in mainland Italy and other world countries. Post-diagnosis relocation of the families seems also unlikely to have occurred, as the excesses were observed in small towns and villages, as well as in the urban areas, where the hospitals with the specialist oncohematology units are located. The information on the commune of residence at the time of diagnosis was missing for 212/4109 patients (5.2%); it seems also unlikely that this might have affected the overall pattern.

Having all the diagnoses reviewed by the same expert hematologist (GB) is certainly an advantage of our study, as this would have spread the error uniformly along the time and over the communes, thus preventing diagnostic bias to affect our findings. Besides, the present study would not have been possible if not by personally collecting and reviewing the incident cases. The validity and completeness of the database we used were previously tested by comparing the results with those based on hospital dismissal records and mortality data in selected areas and time windows [12], and herein confirmed limited to the northern area of the island for the years from 1998 to 2002.

## Conclusions

Our results will hopefully prompt further research on the genetic and environmental determinants of NHL and their interaction, and they might serve as convincing arguments for taking profit of the genetic peculiarities of the Sardinian population by extending cancer registration over the whole island, as planned but not yet implemented. Finally, our study might also help in planning efficiently where to locate the necessary hematology resources to better match the local needs.

## Supporting information

**S1 Data. Excel spreadsheet with the aggregated data of NHL cases incident among the population of the 356 communes of Sardinia Italy in 1974–2003.**
(XLSX)

## Author Contributions

**Conceptualization:** Giorgio Broccia, Cansu Ozsin-Ozler, Pierluigi Cocco.

**Data curation:** Giorgio Broccia, Federico Meloni.

**Formal analysis:** Jonathan Carter.

**Methodology:** Jonathan Carter, Pierluigi Cocco.

**Project administration:** Pierluigi Cocco.

**Supervision:** Sara De Matteis, Pierluigi Cocco.

**Validation:** Giorgio Broccia.

**Visualization:** Cansu Ozsin-Ozler, Federico Meloni.

**Writing – original draft:** Pierluigi Cocco.

**Writing – review & editing:** Giorgio Broccia, Jonathan Carter, Cansu Ozsin-Ozler, Federico Meloni, Sara De Matteis, Pierluigi Cocco.

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
