## [Decision Letter · Decision Letter 0]

6 Jun 2021

PONE-D-21-08974

Incidence of non Hodgkin lymphoma among adults in Sardinia, Italy

PLOS ONE

Dear Dr. Cocco,

Thank you for submitting your manuscript to PLOS ONE. After careful consideration, we feel that it has merit but does not fully meet PLOS ONE’s publication criteria as it currently stands. Therefore, we invite you to submit a revised version of the manuscript that addresses the points raised during the review process by the two Reviewers.

We look forward to receiving your revised manuscript.

Kind regards,

Francesco Bertolini, MD, PhD

Academic Editor

PLOS ONE

Journal Requirements:

2a) If there are ethical or legal restrictions on sharing a de-identified data set, please explain them in detail (e.g., data contain potentially identifying or sensitive patient information) and who has imposed them (e.g., an ethics committee). Please also provide contact information for a data access committee, ethics committee, or other institutional body to which data requests may be sent.

2b) If there are no restrictions, please upload the minimal anonymized data set necessary to replicate your study findings as either Supporting Information files or to a stable, public repository and provide us with the relevant URLs, DOIs, or accession numbers. Please see http://www.bmj.com/content/340/bmj.c181.long for guidelines on how to de-identify and prepare clinical data for publication. For a list of acceptable repositories, please see http://journals.plos.org/plosone/s/data-availability#loc-recommended-repositories.

3. We note that Figure 2 in your submission contain map images which may be copyrighted. All PLOS content is published under the Creative Commons Attribution License (CC BY 4.0), which means that the manuscript, images, and Supporting Information files will be freely available online, and any third party is permitted to access, download, copy, distribute, and use these materials in any way, even commercially, with proper attribution. For these reasons, we cannot publish previously copyrighted maps or satellite images created using proprietary data, such as Google software (Google Maps, Street View, and Earth). For more information, see our copyright guidelines: http://journals.plos.org/plosone/s/licenses-and-copyright.

1.              You may seek permission from the original copyright holder of Figure 4 to publish the content specifically under the CC BY 4.0 license. 

5. Please remove your figures from within your manuscript file, leaving only the individual TIFF/EPS image files, uploaded separately.  These will be automatically included in the reviewers’ PDF.

Reviewers' comments:

Reviewer's Responses to Questions

**Comments to the Author**

1. Is the manuscript technically sound, and do the data support the conclusions?

Reviewer #1: Partly

Reviewer #2: Partly

2. Has the statistical analysis been performed appropriately and rigorously? 

Reviewer #1: I Don't Know

Reviewer #2: Yes

3. Have the authors made all data underlying the findings in their manuscript fully available?

Reviewer #1: Yes

Reviewer #2: Yes

4. Is the manuscript presented in an intelligible fashion and written in standard English?

Reviewer #1: No

Reviewer #2: Yes

5. Review Comments to the Author

Reviewer #1: corrections to English, Non Hodgkin's would be better than Non Hodgkin.

line 88 - only a fraction of cases not but

line 249 - confirms validity not confirm

line 250- and indicates not indicate

line 251 - with what was observed not what observed

Reviewer #2: This manuscript is about a population based study of lymphoma cases diagnosed in Sardinia during a very long time frame. The manuscript is full of observations and is mainly hypothesis generating.

The used approach is interesting and adds value to the work. Nevertheless the study result are severely affected by diagnostic biases caused by the important changes made to the classification of lymphomas during the last 30 years. This problem is only partly hide by the analysis of lymphoma as a whole that represent a second major concern that severely limits the transferability of these data to the clinic. Lymphomas are currently seen as a group of separate entities and every effort should be done to refer data also from cancer based registries to single entities, at least for the most frequent ones. Are there any lymphoma specifica data available at least for the most recent diagnoses, to try to identify if some lymphoma subtype was the cause of different geographical distribution?

A second comment is related to possible temporal changes in the epidemiology of lymphomas during the study period. it would be interesting to analyze this trend separating the study frame in 2 or 3 periods and if number are not to small to identify and explain possible changes.

Finally it would be useful to compare these epidemiological data (i.e. world standardized incidence) with other available from other cancer registries in Italy and to discuss about the comparison (i.e. Luminari et al. Hematologiacl Oncology 2007).

6. PLOS authors have the option to publish the peer review history of their article (what does this mean?). If published, this will include your full peer review and any attached files.

Reviewer #1: **Yes: **Kalyan Saginala

Reviewer #2: No

---

## [Author Response · Author response to Decision Letter 0]

10 Aug 2021

PLoS One Manuscript # PONE-D-21-08974

“Incidence of non-Hodgkin’s lymphoma among adults in Sardinia, Italy”

Response to the Editor

1. “…ensure that your manuscript meets PLOS ONE's style requirements, including those for file naming”

Title page, main text, and file names do now comply with PLOS ONE’s style requirements.

2. “data from this study …available upon request”

We took note of the Journal’s requirement for data availability, and we uploaded the file with the aggregated data as a supplementary file in submitting the revised version.

3. “Figure 2 … contains map images which may be copyrighted”

We contacted the person who created the map of the administrative borders of the Italian communes and made it available on the web site https:\\\\umap.geonue.com/en/map/confini-e-dati-statistici-dei-comuni-ditalia _297#8/40.102/8.973. As reported in the screenshot of the web page we accessed, the map was based on a software created by the National Association of the Italian Communes (ANCITEL) under an Open Database License (ODbL). As reported in the web site https:// opendatacommons.org/licenses/odbl/1-0/index.html, the ODbL license is a license agreement that allows users to freely share, modify, and use under the understanding that the resulting product will also made freely available to the public, and with the condition that the attribution of the original work would be acknowledged to the authors, the product would be shared alike, and the database would be kept open. Such conditions seem to us compatible with the CC BY 4.0 license. Therefore, in the revised version, we added a sentence acknowledging the authors of the map and the creator of the software used for the purpose under the ANCITEL ODbL license. In Submitting the revised manuscript, we also uploaded the screenshot of the web page we accessed (in Italian), along with its translation to English, as an “other” file.

4. The “… ethics statement should only appear in the Methods section …”

The ethics statement is now the last paragraph of the Methods section.

5. “…remove your figures from within …[the]… manuscript file, leaving only the individual TIFF/EPS image files, uploaded separately.” 

We removed Fig 2 from the manuscript file, and uploaded separately its parts as individual TIFF image files.

Response to the reviewers’ comments

Reviewer 1. Grammatical errors.

We are grateful to the reviewer for picking some grammatical errors in the text. We amended those and a few others. However, the plural form of the verbs “to confirm” (line 249) and “to indicate” (line 250) is correct as the subject (Cancer Registry data) is plural.

Reviewer 2.

1. “…diagnostic biases caused by the important changes made to the classification of lymphomas during the last 30 years”.

We dedicated an entire paragraph, the first under the subheading “Limitations” from line 309 to 328 on page 10 to discuss this point. As we explain in the text, having the first author, himself a senior hematologist, collecting and revising each diagnosis of the incident hematological malignancies in 1974-2003 should reassure us and the readers about diagnostic bias due to the changes in the lymphoma classification that occurred along the study period.

2. ”…the analysis of lymphoma as a whole …[is]… a second major concern that severely limits the transferability of these data to the clinic. Lymphomas are currently seen as a group of separate entities and every effort should be done to refer data also from cancer based registries to single entities”. “Are there any lymphoma specific data available at least for the most recent diagnoses, to try to identify if some lymphoma subtype was the cause of different geographical distribution?”.

We definitely agree with the reviewer that every effort should be addressed to explore individual subtypes. However, the point is that the epidemiological analysis of time trends requires using a uniform definition of the disease entity across the time span object of the investigation. This imposes keeping using the initial definition. For the same reason, the IARC CI5 volumes and the Cancer Registries also keep using the NHL definition. A few studies have explored time trends, although within a still limited time frame (Morton L et al. Blood 2006;107(1):265-276; Luminari S et al. Hematol Oncol 2007;25:189-197; van Leeuwen MT et al. Int J Cancer 2014;135(9):2146-2156; Lim RB et al. Int J Cancer. 2015;137(11):2674-2687) or survival from (Al-Hamadani M et al. Am J Hematol 2015;90: 790–795; Ye X et al. BMJ Open 2017;7(7):e015106), major B-cell lymphoma subtypes. To the best of our knowledge, enough information might be currently available to explore the spatial distribution of lymphoma subtypes as well. We hope that for the future IARC would promote the registration of lymphoma subtypes besides keeping the old NHL definition, so to make possible what the reviewer proposes at a larger scale. 

2. “…possible temporal changes in the epidemiology of lymphomas during the study period. it would be interesting to analyze this trend separating the study frame in 2 or 3 periods and if number are not too small to identify and explain possible changes.”

We are grateful to the reviewer for raising this interesting comment. To identify any joinpoint in the regression of NHL incidence against time, we plotted the residuals against the values predicted by the regression equation. As shown in figure 1, high or low residuals did not cluster in a specific range of predicted values. Therefore, we interpreted the upward trend we observed as continuous, with no evidence of a sudden change in slope, which might have been related to introduction of new diagnostic tests, or upgrade or new opening of onco-haematology facilities, or any acute environmental event. We added a new sentence at the end of the Time trend in NHL incidence paragraph of the Results section.

Figure 1. Residuals vs values predicted by the regression equation of 1974-2003 NHL incidence in Sardinia.

4. “… compare these epidemiological data (i.e. world standardized incidence) with other available from other cancer registries in Italy and to discuss about the comparison (i.e. Luminari et al. Hematol Oncol 2007).”

We again grateful to the reviewer for its useful suggestion. We retrieved the paper by Luminari et al., we extracted the corresponding figures from our data set, and we compared the age standardized incidence rates (World standard population) in males and females for the corresponding period. Rates from the Modena Cancer Registry were substantially higher in both genders. In the revised version, we added a paragraph under the subheading “Time trend in NHL incidence” to present and comment about such comparison.

---

## [Decision Letter · Decision Letter 1]

27 Sep 2021

PONE-D-21-08974R1Incidence of non-Hodgkin's lymphoma among adults in Sardinia, ItalyPLOS ONE

Dear Dr. Cocco,

Thank you for submitting your manuscript to PLOS ONE. After careful consideration, we feel that it has merit but does not fully meet PLOS ONE’s publication criteria as it currently stands. Therefore, we invite you to submit a revised version of the manuscript that addresses the points raised during the review process by Reviewer #1.

We look forward to receiving your revised manuscript.

Kind regards,

Francesco Bertolini, MD, PhD

Academic Editor

PLOS ONE

Journal Requirements:

Additional Editor Comments (if provided):

Reviewers' comments:

Reviewer's Responses to Questions

**Comments to the Author**

1. If the authors have adequately addressed your comments raised in a previous round of review and you feel that this manuscript is now acceptable for publication, you may indicate that here to bypass the “Comments to the Author” section, enter your conflict of interest statement in the “Confidential to Editor” section, and submit your "Accept" recommendation.

Reviewer #1: All comments have been addressed

Reviewer #2: All comments have been addressed

2. Is the manuscript technically sound, and do the data support the conclusions?

Reviewer #1: Yes

Reviewer #2: Yes

3. Has the statistical analysis been performed appropriately and rigorously? 

Reviewer #1: Yes

Reviewer #2: Yes

4. Have the authors made all data underlying the findings in their manuscript fully available?

Reviewer #1: Yes

Reviewer #2: Yes

5. Is the manuscript presented in an intelligible fashion and written in standard English?

Reviewer #1: Yes

Reviewer #2: Yes

6. Review Comments to the Author

Reviewer #1: 38th page - suggests not suggest

42 page - the communes associated with posterior probability are located on the extreme right of the incidence distribution curve of the 356 communes in the region would be better way to phrase and write than the current original one

Reviewer #2: (No Response)

7. PLOS authors have the option to publish the peer review history of their article (what does this mean?). If published, this will include your full peer review and any attached files.

Reviewer #1: **Yes: **Kalyan Saginala

Reviewer #2: **Yes: **Stefano Luminari

---

## [Author Response · Author response to Decision Letter 1]

8 Oct 2021

Reviewer 1. 

1. Grammatical error (“38th page - suggests not suggest”).

The plural form of the present tense of the verb “to suggest” is used twice: one in the abstract (line 10) and one on page 10 of the manuscript (last line). In both instances, the plural form is correct as the subject (Cancer Registry data) is plural.

2. “42 page - the communes associated with posterior probability are located on the extreme right of the incidence distribution curve of the 356 communes in the region would be better way to phrase and write than the current original one”

We rephrased the statements that describe the location of high incidence communes in the graph and in the map as it follows: “Another 22 communes are located on the extreme right of the incidence distribution curve of the 356 communes in the region, with a posterior probability of an excess NHL incidence in the total population ranging 80 – 94%. On the map, these communes tended to concentrate in the northern (20/35 communes) and central (12/35) areas of the island, with only three in southern Sardinia, including the major urban area of Cagliari.” We are grateful for the reviewer suggestion, and we hope the sentence sounds better.

---

## [Decision Letter · Decision Letter 2]

3 Nov 2021

Incidence of non-Hodgkin's lymphoma among adults in Sardinia, Italy

PONE-D-21-08974R2

Dear Dr. Cocco,

We’re pleased to inform you that your manuscript has been judged scientifically suitable for publication and will be formally accepted for publication once it meets all outstanding technical requirements.

Kind regards,

Francesco Bertolini, MD, PhD

Academic Editor

PLOS ONE

Additional Editor Comments (optional):

Reviewers' comments:

Reviewer's Responses to Questions

**Comments to the Author**

1. If the authors have adequately addressed your comments raised in a previous round of review and you feel that this manuscript is now acceptable for publication, you may indicate that here to bypass the “Comments to the Author” section, enter your conflict of interest statement in the “Confidential to Editor” section, and submit your "Accept" recommendation.

Reviewer #1: All comments have been addressed

2. Is the manuscript technically sound, and do the data support the conclusions?

Reviewer #1: Partly

3. Has the statistical analysis been performed appropriately and rigorously? 

Reviewer #1: Yes

4. Have the authors made all data underlying the findings in their manuscript fully available?

Reviewer #1: Yes

5. Is the manuscript presented in an intelligible fashion and written in standard English?

Reviewer #1: Yes

6. Review Comments to the Author

Reviewer #1: interesting article, very well summarized the epidemiological background and incidence of NHL among adults in Sardinia, Italy, title is appropriate

7. PLOS authors have the option to publish the peer review history of their article (what does this mean?). If published, this will include your full peer review and any attached files.

Reviewer #1: **Yes: **kalyansaginala

---

## [Editor Report · Acceptance letter]

21 Jan 2022

PONE-D-21-08974R2 

*Incidence of non-Hodgkin’s lymphoma among adults in Sardinia, Italy*

Dear Dr. Cocco:

I'm pleased to inform you that your manuscript has been deemed suitable for publication in PLOS ONE. Congratulations! Your manuscript is now with our production department. 

Kind regards, 

on behalf of

Dr. Francesco Bertolini 

Academic Editor

PLOS ONE